# Consumer Preferences for Craft Beer by Means of Artificial Intelligence: Are Italian Producers Doing Well?

Vittoria Pilone [1] , Alessandro Di Pasquale [2] and Antonio Stasi [2,*]

1 Department of Economy, Management and Territory (DEMeT), University of Foggia, 71121 Foggia, Italy
2 Department of Agriculture, Food Natural Resources and Engineering (DAFNE), University of Foggia, 71121 Foggia, Italy
* Correspondence: antonio.stasi@unifg.it

**Abstract:** To identify the key drivers of consumption, we analyzed consumer preferences and estimated the willingness to pay for craft beer compared with industrial products in Italy. For this purpose, we conducted an ad hoc survey of 469 craft beer drinkers and set up an econometric strategy using a machine learning estimation technique. The main results show that young consumers, the ability to perceive and evaluate quality, and the frequency of consumption are the main profile elements that, more than others, orient preferences. In the meantime, sustaining local beer producers, sharing good time with friends, and the perception of beer as healthier compared with other drinks are also important. The most preferable product attributes are can packaging and the search for unique taste.

**Keywords:** consumer behavior; willingness to pay; artificial intelligence; marketing strategies; Italy

## 1. Introduction

Italy never had a traditional brewing culture; nevertheless, from 1996, the craft beer sector has grown substantially. In fact, during more recent years, this production has eroded market shares to wine consumption, finding new opportunities within young target consumers. This positive trend is also supported by the frequent eating-out habits of Italian consumers [1–3]. The contemporary Italian craft beer sector represents a real market niche, numbering over 1000 breweries and beer firms characterized by local and small-scale enterprises, an annual production higher than 590,000 hl, and a market share of around 3% of the total Italian beer market [4,5]. For this product, the most important distribution channels are represented by hotels, restaurants, and the catering industry [2].

In this market arena, competitiveness has grown, and brewers are seeking new opportunities and marketing strategies, oriented to new consumer targets, and mainly centered on persuasive communication, innovative packaging, and brand identity strengthening [6–8].

With the purpose of supporting producer marketing strategies, this study investigated new consumer markets, with the particular aims of:

Obj. 1—Identifying the main drivers of craft beer consumption within Italian consumers;

Obj. 2—Verifying whether the current marketing and communication strategies adopted by Italian breweries are coherent with new consumers needs and wants.

Based on the obtained outcomes, this study may offer new marketing pathways for breweries to succeed.

On this topic, the economic literature already presents a number of studies, and, in more recent years, several authors have proposed different contributions. Some of these are focused on new consumer trends, consumer characteristics, consumption habits, attitudes, and motivations, considered as drivers for preferences [2,6,9–20], while other studies are centered on positive impact of this sector to develop specific rural areas and to improve small producers' competitiveness [1,12,21–30].

Nevertheless, at present, to the best of our knowledge, no studies have compared consumer preferences analysis with producers' marketing and communication strategies to provide real suggestions for orienting a company's focus.

To fill this gap, in this paper, an analysis on consumer preference and an explorative study on the communication strategies of craft beer enterprises are presented. In particular, more relevant consumer characteristics and product attributes are detected and evaluated, with the aim of identifying the most significant drivers orienting craft beer choices and its consumption. For these aims, random forest regression was used to find the principal drivers orienting preferences, whereas stepwise OLS regression and a post-lasso OLS regression were used to estimate WTP and to confirm the most valuing factors. The results of this analysis are compared with an explorative investigation of the communication strategies of craft beer enterprises on social networks.

## 2. Background

### 2.1. Literature Review

Market opportunities of the craft beer sector have determined a simultaneous growing interest towards empirical and scientific analyses in all the related subjects.

A large body of this literature is focused on microbiological quality, safety, and innovation in the craft brewing process [31–39]; other studies are related to economic research. Most studies on this topic discuss consumer characteristics, consumption habits, attitudes and motivations, and preferred product attributes, exhibiting agreement about new consumer trends in craft beer market. In general, new consumers are young and regular drinkers, with high levels of education, a strong identity, and an ability to recognize craft beer quality. This consumer profile has been confirmed in different studies conducted in a range of countries [16,17,19]. In particular, in their survey carried out in Brazil, Carvalho et al. [17] highlighted differences between traditional and innovative consumers based on age, education, and income; Muggah and McSweeney [19] identified that unique craft beer taste is the most important attribute for Canadian consumers' choice, also emphasizing some differences between males and females. In addition, in comparison studies between countries such as Germany and Canada, it was found that the consumers of craft beer are mainly male; however, in Italy, women are willing to consume this product as well [15,19].

With a focus on Italian consumption, several authors have investigated socio-demographic characteristics and consumer habits, showing that Italian consumers can be clustered in different segments having the knowledge and awareness of product quality in common [2,6,9–11,13,16,18,20]. In particular, Rivaroli et al. [16] and Aquilani et al. [20] found that a young age and frequent consumption are the main features characterizing new Italian consumers. Crociata [11] highlighted that Italian consumption is both individual and associated with socio-cultural occasions; however, with reference to attitudes and motivations, Lerro et al. [13] and Spadoni et al. [2] agree that drivers for preferences researched in Italian consumer needs are sharing friendly moments or tasting high-quality products. In more recent studies, unique taste, technology, authenticity, color, and supporting small and local producers are considered to be the most important product attributes and the main drivers orienting choices. Conversely, the high quality of raw materials, geographical origin, and product price are generally less significant [6,9,12,13]. Focusing on the packaging, although traditional glass bottles have historically been chosen [18], younger and new craft beer consumers have declared preferring can packaging [6].

Assessing the impact of the craft beer sector for the development of specific rural areas and to improve the competitiveness of small producers, the authors agree that the expansion of craft breweries has positively contributed to the development of the Italian beer sector.

Initially, the international integration of consumer trends encouraged the knowledge of beer typologies and styles among Italian people. These changes fostered small and specialized firms to enter the craft beer market. Subsequently, legitimization and emulation

effects played a key role in sustaining the diffusion and rise of these new breweries [1]. Over the years, the nature and the local orientation of Italian craft breweries have been considered key aspects of their success because they can satisfy consumers' desires to re-establish a connection to local places, communities, and economies [27]. These immaterial attributes, not appliable in mass-produced beer, have given Italian craft beer a strong identity and contributed to the development of specific local areas, improving the competitiveness of small-scale producers [25,26,30].

However, other studies have demonstrated the limited relevance of geographical and local factors for new entering companies. In these cases, market power and individual features should be considered as more effective to define competitive strategies [24]. This perspective can also explain the international interest of large companies to invest in the microbrewing industry, rather than observing microbreweries enter in related businesses. This evolving scenario confirms the importance of differentiating production from competitors [23].

More recent studies have reported a sort of segmentation in Italian breweries' strategies. Although some of them emphasize agricultural phase and impact, others are more prone to emphasize product attributes and brand reputation. Moreover, other breweries stress the importance of sustainability-connected elements to promote their production [21,22,29].

These key aspects can be considered as interesting for the further development of this sector and to orient marketing and communication strategies aimed to attract new consumer segments.

### 2.2. The Craft Beer Sector in Italy

Based on the American craft beer revolution, this sector started in Italy in 1996, when the first small breweries started producing [28,40]. Over the years, the product has developed in a market led mainly by standardized beer.

Initially, the unique taste and the different beer styles stimulated consumers' curiosity. Afterwards, as a real outsider of industrial beer, Italian consumers have increasingly appreciated this no conventional product.

Nevertheless, product identity was not clear until July 2016, when, with law no.154, articles no. 35 and 36, the Italian Ministry of Agricultural, Food and Forestry Policies provided the first legal craft beer definition. Based on this, in Italy, differently from other countries, a brewery can be identified as "craft" if some important requests are satisfied. A brewery has to produce no more than 200,000 hl/year; it depends legally and economically on other companies; and its beer is produced without pasteurization and microfiltration processes.

This regulation clarified the main differences between industrial and craft beer based on technological and economic aspects. In addition, it represents a legal tool for protecting small and medium breweries with limited production.

In only a few years, the number of producers has grown exponentially; at present, the sector has approximately 1600 active companies, including breweries, brewpubs and beer-firms, with an annual production of 590,000 hl, [4,5]. Craft beer has a market share higher than 3% of the total beer sector; therefore, it can be considered as a real niche market in terms of continuous expansion.

Table 1 shows the distribution of breweries in Italy. Specifically, based on Nielsen areas[1], Table 1 indicates that craft beer producers are evenly distributed in Italy, with a presence of about 400 companies per each area, as specified in Table 1. This aspect can be considered relevant because it highlights that the development of the craft beer sector has been similar in each area across Italy; consequently, new hints for innovative marketing and communication strategies could be useful for increasing the competitiveness of this sector all over the country.

**Table 1.** Distribution of breweries in Italy based on region and Nielsen area.

| | Region | | Number of Breweries |
|---|---|---|---|
| Area 1 | Liguria | | 34 |
| | Lombardia | | 245 |
| | Piemonte | | 120 |
| | Valle D'aosta | | 10 |
| | | Tot. | 409 |
| Area 2 | Emilia-Romagna | | 118 |
| | Friuli Venezia Giulia | | 51 |
| | Trentino Alto-Adige | | 55 |
| | Veneto | | 159 |
| | | Tot. | 383 |
| Area 3 | Lazio | | 113 |
| | Marche | | 75 |
| | Sardegna | | 53 |
| | Toscana | | 118 |
| | Umbria | | 37 |
| | | Tot. | 396 |
| Area 4 | Abruzzo | | 56 |
| | Basilicata | | 26 |
| | Calabria | | 37 |
| | Campania | | 99 |
| | Molise | | 15 |
| | Puglia | | 92 |
| | Sicilia | | 80 |
| | | Tot. | 405 |
| Italy | Tot. | | 1593 |

Source: own elaboration on data from Microbirrifici.org [5].

## 3. Materials and Methods

### 3.1. The Methodology: The Base Price Anchoring Contingent Valuation Method

In this study, we conducted a demand analysis by estimating the effect of craft beer attributes on consumers' willingness to pay (WTP). Moreover, we measured the impact of segmentation variables on demand.

The methodological approach we employed for eliciting the value that consumers assign to innovative products and services was the contingent evaluation method (hereafter, CVM). In fact, we conducted an original evaluation study based on primary data collected among Italian beer consumers with an online questionnaire survey.

We used a personalized version of the open-ended (OE) elicitation format of WTP. The innovative element of our methodology is given by anchoring the elicitation of the WTP to the price of the basic product; in our experiment, this was industrial beer. This approach is hereafter referred to as base price anchoring (BPA) CVM.

We referred to OE for several reasons:

1. The format yields easy and fast answers, which gives room for other in-depth questions to explore preferences and habits in an online survey, which is expected to have a more limited attention spam from respondents [41];
2. The general tendency found in the literature is that the open-ended format results in lower WTP estimates than the closed-ended format when used in a hypothetical setting [42,43];
3. In the literature, the presence of experimental results comparing hypothetical and actual WTP shows that the hypothetical bias is not higher, or even lower, for the open-ended format compared with the closed-ended format [44,45];
4. There is no incentive to overstate WTP, nor an incentive to state zero WTP.

Certainly, OE elicitation of WTP for an unconventional product in each market, such as craft beer in Italy, could leave consumers doubtful because they may eventually not have clarity on the magnitude of market price. Therefore, we first asked respondents to state the last price they paid for the basic product, i.e., industrial beer.

Having stated the anchoring price, the respondent was then asked to state their maximum WTP for the described craft beer product.

### 3.2. Empirical Analysis and Data Collection

To achieve the research goals, the empirical analysis was articulated in two parts: an investigation on consumer preferences towards new craft beer products, and an exploratory study focused on current marketing and communication strategies adopted by producers.

The demand analysis aimed to identify the most significant product characteristics and consumer drivers able to orient consumer preferences and to estimate the willingness to pay for them. This was conducted by a survey across Italy, based on declared consumer preferences on craft beer followed by the implementation of different statistic models. Random forest regression was useful to identify key drivers orienting preferences, while stepwise OLS regression and post-lasso OLS regression were used to estimate WTPs and to confirm the most valuing factors.

Data collection was carried out through an online questionnaire distributed using Google Forms, from September to December, 2020. The survey involved 469 Italian consumers of craft beer, who declared to have drunk craft beer a minimum of once in the last year.

The questionnaire was composed of different questions clustered into three different sections. The first section of 15 questions collected information concerning socio-demographic characteristics of the sample. The second section of 20 questions aimed to examine interviewees' purchase behavior and consumption habits. In this part of questionnaire, the perceptions of two different packages were identified. Respondents were asked to evaluate specific communication aspects of two beer packaging media, comparing one glass bottle and one aluminum can. In the third section of the questionnaire, comprising 15 questions, the motivations and attitudes of consumers were surveyed. The questionnaire was organized in exhaustive close-ended questions to simplify the response. In general, the questionnaire was intended to identify information in 10 min about the consumers' knowledge and awareness of the investigated product. In addition, purchasing behaviors and preferences, as well as the key reasons supporting craft beer consumption, were examined. Moreover, and according to the literature, questions about taste, beer style, healthiness, packaging communication aspects, concepts related to sustainability and conviviality, were presented to respondents [21,26].

With regard to marketing and communication strategies adopted by craft beer producers in Italy, an exploratory analysis was carried out, based on secondary data obtained by the most recent annual report, AssoBirra [4]. In addition, in-depth research was conducted, concerning key web-based communication strategies used by Italian craft breweries. These aspects were considered as relevant because of their ability to highlight breweries' vitality and involvement in innovative marketing strategies. Thus, from all active breweries in Italy, a sample of 100 enterprises, stratified by geographical area, was randomly extracted, and four main aspects were examined. Specifically, the number of monthly activities on Facebook and Instagram social networks, the presence of a company website, and active e-commerce were evaluated. Data collection was conducted from January to February, 2021.

### 3.3. Econometric Approach and Analysis

Having the maximum WTP obtained through BPA CVM questions, as well a set of variables concerning preferences and personal data, regression analysis could be carried out to estimate the impact of those variables on WTP.

No zero WTP was present in the database; therefore, there was no need for truncated regression analysis.

Given such a premise, we decided to explore alternative approaches to standard econometric analysis and implement a new methodology adopting artificial intelligence regression analysis (hereafter, AI) and machine learning (ML) techniques.

### 3.3.1. Artificial Intelligence and Machine Learning

In this section, we address the concept of unsupervised learning, which embraces models such as support vector machine, neural networks, and tree-based models, used for both regression and classification problems.

According to the statistical learning theory (SLT), the problem of supervised learning is formulated as follows. Given a set of training data $D = \{(x_1, y_1) \ldots (x_1, y_1)\}$ in $R^n \times R$, sampled according to unknown probability distribution $P(x, y)$, and a loss function $L(y, f(x))$ that measures the error when, for a given $x, f(x)$ is "predicted" instead of the actual value, *y*.

It is worth pointing out that there is no information on the underlying joint probability functions. Therefore, it is necessary to perform a "distribution-free" approach, where the only information available is a training dataset.

The problem consists of finding a function, f, that minimizes the expectation of the error in new data, i.e., to find a function, f, that minimizes the expected error:

$$\int L(y, f(x)) P(x, y) dx dy \tag{1}$$

$P(x, y)$ is unknown; therefore, we need to use an induction principle to infer from the one available training example as a function that minimizes the expected error. The principle used was empirical risk minimization (ERM) over a set of possible functions, called hypothesis space. Formally, this can be written as minimizing the empirical error:

$$\frac{1}{l} \sum_{i=1}^{l} L(y_i, f(x_i)) \tag{2}$$

Machine learning tools are the so-called "nonparametric" models. "Nonparametric" does not mean that the ML models do not have parameters at all. In contrast, their "learning" is the crucial issue here; indeed, unlike in classic statistical inference, the parameters are not predefined, and their number depends on the training data used.

### 3.3.2. Random Forest Regression Models

Regression trees usually produce low-bias and high-variance estimations; therefore, they are a good candidate for ensemble methods. Indeed, random forests basically consist of building an ensemble of decision trees grown from a randomized variant of the tree; this method is useful to determine the error reduction, decreasing the prediction variance, and preserving the bias. Starting from a single learning set, the basic idea is to introduce a random perturbation into the learning procedure to introduce a differentiation among the trees and combine the predictions of all these trees using aggregation techniques.

Breiman, in a study published in 1996 [46], proposed a first aggregation method, so-called bagging, in which different trees are built using random bootstrap copies of the original data. Its natural evolution, random forest, was developed by Breiman in 2001 [47]. In random forest approaches, bagging is extended and combined with the randomization of input variables that are used when considering candidate variables to split internal nodes, *t*. Instead of looking for the best split, $s^*$, among all variables, the algorithm chooses a random subset of *K* variables for each node and then determines the best split using these variables.

### 3.3.3. Model Training

In this section, we describe the training phase procedure. The dataset was randomly split into a training set and an independent test set, containing 70% and 30% of the total data, respectively. Machine learning model training involves a procedure of hyperparameter

fine-tuning until the model is optimized. Specifically, we used 10-fold cross-validation, which means that the chosen 70% of the sample is further randomly partitioned into ten subsamples of equal size. Thus, nine subsamples were used as training data, and the remaining subsample was used for validation. The cross-validation process was repeated ten times (folds) such that each of the subsamples was used only once for validation. The hyperparameters were updated based on the average of ten results. The learning phase was complete when the hyperparameters were optimal, producing minimum prediction errors; then, we tested the predictive performance of the final model using the unobserved 30% test sample. Random forests require few specifications during the training phases. The process consists of feeding the model the training set and subsequently assessing its accuracy.

## 4. Results and Discussion

### 4.1. Characteristics of the Sample

The sample consisted of 469 Italian consumers, distributed geographically as follows: Northern Italy, 38%; Central Italy, 20%; and Southern Italy, 42%. As described in Table 2, the sample was mainly constituted of males (71%), with an average age of 34, but ranging from 18 to 86 years old. There were 34% of the respondents employed in both the public and private sector, whereas 24% were students; 50% had a college degree, and 46% had a high school diploma. Family income ranged between EUR 1000 and EUR 3000.

**Table 2.** Characteristics of the sample.

| Variable | Category | % | Media | Dev. St. | Min | Max |
|---|---|---|---|---|---|---|
| Gender | Male | 71 | | | | |
| | Female | 29 | | | | |
| Age | | | 34 | 11.43 | 18 | 86 |
| Education | Primary and secondary school diploma | 4 | | | | |
| | High school diploma | 50 | | | | |
| | Bachelor's degree or higher leve | 46 | | | | |
| JOB | Out of the market (housewife, unemployment, retirees) | 7 | | | | |
| | Student | 24 | | | | |
| | Worker | 13 | | | | |
| | Employee | 34 | | | | |
| | Executive | 21 | | | | |
| Monthly household income | <EUR 1000 | 8 | | | | |
| | EUR 1001–3000 | 54 | | | | |
| | EUR 3001–5000 | 20 | | | | |
| | >EUR 5001 | 18 | | | | |
| Geographical residence | North | 38 | | | | |
| | Center | 20 | | | | |
| | South and Islands | 42 | | | | |
| | Out of Italy | 1 | | | | |

As shown in Table 3, craft beer was purchased from ho.re.ca. (58% of cases) or from retail stores (42%). E-commerce represents an emerging channel for the sales, while retail and wholesale represent a minimal share of sales.

**Table 3.** Consumer behaviors of the sample.

| Variable | Category | % |
|---|---|---|
| Purchase channel | Ho.re.ca | 58 |
| | Retail | 42 |
| Retail store (multiple choice) | Local brewery | 55 |
| | Beer shop/pub (for retail) | 45 |
| | E-commerce | 29 |
| | Retail and wholesale | 7 |
| | Homebrewing | 5 |
| Frequency of consumption | >4 times a week | 15 |
| | 2–4 times a week | 19 |
| | 1–2 times a week | 35 |
| | Around 1 time a week | 20 |
| | <1 time a month | 11 |
| Consumption occasion | During main meals | 32 |
| | Away from main meals | 68 |
| Knowledge Source (multiple choice) | Journals | 15 |
| | Social media | 35 |
| | Word of mouth | 42 |
| | Ho.re.ca menu | 52 |
| | Personal interest | 13 |

When consumed at home (55% of cases), craft beer is generally bought from local breweries, which supply craft beer to consumers, supporting the local economy; otherwise, it is drunk in pubs and beer shops. Craft beer was consumed once a week or more in 70% of cases. The time of the day was generally away from the main meal (68%).

Information and communication in the craft beer sector has its specificities. Pub menus are the main tool to gain information on craft beer brands (52% of cases), as well as word of mouth (42% of cases). In the meantime, social networks represent a new and effective tool to engage with consumers and gain popularity of the brand (35% of cases), together with personal interest and enthusiasm for the product. Other sources are negligible.

The main consumer motivations are described in Table 4; for the different motivations investigated, a color scale ranging from green to red, indicating decreases in the percentage consensus, emphasizing that Italian people choose craft beer for some specific reasons. Firstly, it is valued for its unique taste compared with industrial products, and because it is perceived to be healthier than other alcoholic drinks. Secondly, choosing craft beer is associated with the enjoyment of evaluating its quality, as well as supporting small and local producers. Among the options suggested in the questionnaire, the trendy association of craft beer is not a motivator of choice.

*4.2. Consumers' Profile and Willingness to Pay*

The aim of the analysis was to determine the willingness to pay for artisanal beer, comparing the price with that of industrial beer, and exploring the consumers' characteristics influencing the consumers' attitude towards the price, and investigating their profile.

To answer the questions, artificial intelligence estimation methods have been applied, more specifically, random forest and a post-lasso regression model set to a contingent evaluation model specification.

**Table 4.** Consumer motivations.

| | Strongly Disagree | Disagree | Somewhat Disagree | Neither Agree /Nor Disagree | Somewhat Agree | Agree | Strongly Agree |
|---|---|---|---|---|---|---|---|
| I consume craft beer for its unique taste. | 3% | 6% | 3% | 4% | 12% | 29% | 43% |
| I believe craft beer to be healthier than other alcoholic drinks. | 4% | 0% | 7% | 19% | 17% | 33% | 21% |
| I consume craft beer to share a good time with my friends. | 9% | 7% | 7% | 31% | 17% | 17% | 12% |
| I consume craft beer because it is trendy. | 67% | 15% | 10% | 5% | 3% | 1% | 0% |
| I consume craft beer to support small and local producers. | 5% | 6% | 6% | 15% | 25% | 25% | 18% |
| I am able to evaluate craft beer quality based on its technology and its style. | 6% | 9% | 8% | 13% | 22% | 26% | 16% |

Colors from red to green is a visual information about the frequency of consumers checking the given option.

Table 5 reports the random forest regression results. Table 6 reports the regression results of the lasso and post-lasso regression. In addition, for a robustness check, stepwise regression analysis results have also been reported.

**Table 5.** Results from the random forest regression analysis.

| Variable | Importance |
|---|---|
| Age | 100.000 |
| Perception of quality | 78.733 |
| Last price paid for beer | 76.128 |
| Frequency of consumption artisanal beer | 58.893 |
| Small business support as motivation | 43.939 |
| Sharing | 37.055 |
| Healthy | 36.074 |
| Glass bottle | 29.941 |
| Taste | 29.601 |
| Income | 28.684 |
| Study/job/personal interest | 28.593 |
| Household | 20.196 |
| Trendy | 19.176 |
| Drinking with meal | 14.199 |
| Education | 13.580 |
| Color | 9.806 |
| Breweries as purchase criteria | 9.630 |
| With friends | 9.616 |
| Brand ID | 9.605 |
| Social | 9.262 |
| Root Mean Square Error | 0.316 |
| Mean Absolute Error | 0.258 |

**Table 6.** Results from the stepwise OLS regression and post-lasso OLS regression analyses.

| Variable | LASSO | Post-Lasso OLS | Stepwise OLS |
|---|---|---|---|
| freq_artisanal | 0.034 | 0.057 | 0.053 ** [0.015] |
| smallbusiness | | | −0.015 [0.01] |
| age | −0.001 | −0.005 | −0.006 ** [0.001] |
| healthy | | | −0.015 [0.011] |
| at_pub | | | −0.042 [0.031] |
| gender | | | −0.078 * [0.033] |
| sharing | | | 0.018 + [0.01] |
| ecommerce | | | −0.059 + [0.034] |
| quality | 0.014 | 0.021 | 0.015 [0.01] |
| color | | | −0.059 + [0.033] |
| glass_pack | −0.063 | −0.105 | −0.093 ** [0.035] |
| personal_interest | 0.074 | 0.162 | 0.178 ** [0.045] |
| taste | 0.010 | 0.025 | 0.028 ** [0.01] |
| Constant | 0.437 | 0.417 | 0.686 ** [0.116] |
| Lambda | 35.555 | | |
| Observations | | | 469 |
| R-squared | | 0.261 | 0.238 |

standard errors in brackets
** $p < 0.01$, * $p < 0.05$, + $p < 0.1$

Table 5 and Figure 1 report the level of importance of the variables in determining the price that consumers are willing to pay for artisanal beer, in addition to what they already pay for industrially produced beer. As shown by the results, machine learning techniques do not enable standard errors to be calculated; thus, no t-statistics and *p*-value for the significance are indicated, compared with the regression results.

Table 6 reports the significant coefficients in predicting the WTP level. Age was the most relevant consumer characteristic affecting the WTP. Quality perception, as well as the frequency of consumption, also represented very important characteristics. Secondly, other variables with interesting impacts on marketing strategies and communication were helping small businesses, consuming the product in a convivial environment with friends, and the type of packaging. Lasso and post-lasso regression analyses, in addition to random forest results, helped us to understand the direction of the impact under the linear hypothesis. In fact, age was slightly negatively related to the willingness to pay. However, when estimating the WTP density of age, as shown in Figure 2c, consumers between 20 and 30 years old were those expressing the highest WTP, although one may have expected a contrasting result. Nonetheless, the largest frequency of consumers was positioned on a

WPT which ranged between EUR 0.5 and EUR 1, for those aged 30 years old. Another relevant result concerned the perception and ability of understanding quality and WTP. A positive correlation has been confirmed, as shown in Figure 2b, indicating that most consumers and those demonstrating a high WTP are positioned have a high level of understanding the quality of craft beer.

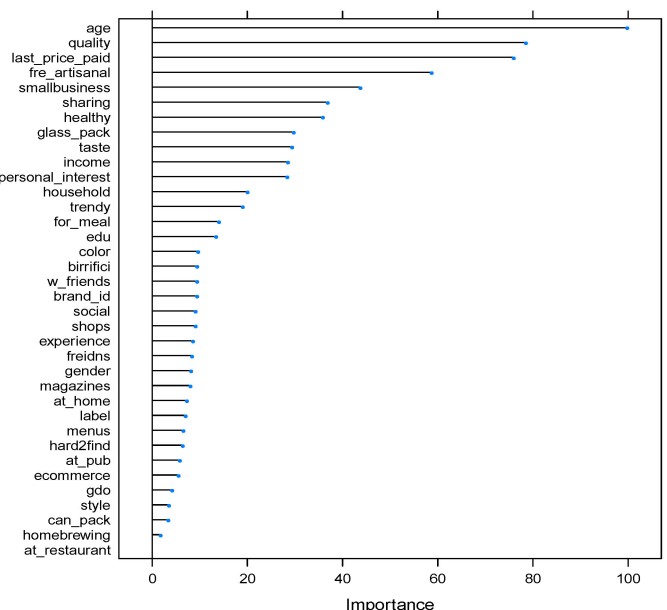

**Figure 1.** Random forest estimation of attribute importance. Representation of the results.

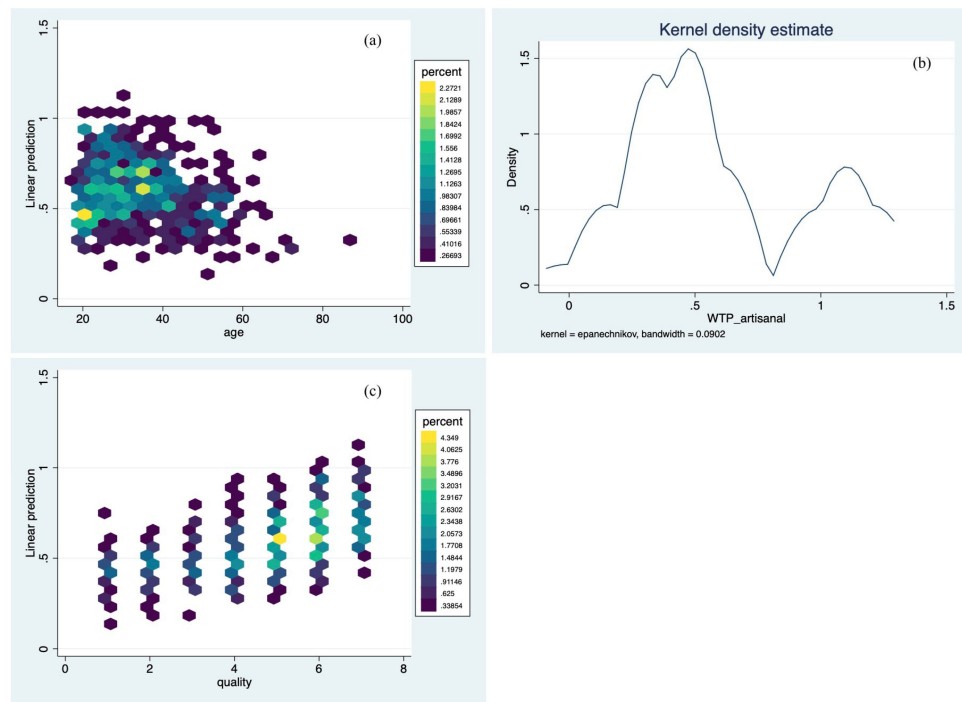

**Figure 2.** Density of WTP. (**a**) density of the WTP on age; (**b**) Frequency distribution of the WTP for craft beer; (**c**) density of the WTP on quality.

In general, as shown in Figure 2a, WTP exhibited a very specific frequency distribution. There was a large share of consumers positioned at approximately EUR 0.50, and another share of the sample with a WTP higher than EUR 1.00. The market, therefore, appears to be

split in two segments: the experts, those able to distinguish and comment on quality; and youngsters, those younger and with a high willingness to pay.

The outcome analyses on consumer preferences confirm and reinforce previous results by other studies conducted in the European market. In particular, in their study aiming to compare the "purely" commercial beer consumer profile with that of commercial beer consumers who had already tasted craft beers, Aquilani et al. (2015) [20] observed that the aroma and perceived quality, a preference for draft beer, and drinking beer frequently were factors that explained the propensity of industrial beer drinkers to select craft beer. It was also found that beer consumers' evaluations of characteristics and brands differed depending on whether they had previously tasted craft beer or not. Moreover, craft beer was chosen according to different flavor preferences compared with commercial beer; it is mainly drunk by frequent beer drinkers in pubs and with family members, and it is perceived to be of higher quality than commercial beer due to the raw materials used for brewing and its overall quality.

### 4.3. Results of the Beer Packaging Comparison Study

Among the variables determining the willingness to pay for artisanal beer, packaging, i.e., glass or can, was found to be significant. Beer packaging, in fact, is one of the most valuable vehicles of information and values [8]. To follow up the consumer profile and willingness to pay analysis, a more accurate focus on packaging was conducted, based on the survey responses.

More specifically, the perception of the product based on the packaging was explored throughout different dimensions: traditional vs. trendy; expensive vs. cheap; sad vs. happy; old vs. young; common vs. sophisticated; unpleasant vs. pleasant; not socializing vs. socializing; poor quality vs. high quality; local vs. global; non-natural vs. natural; and conventional vs. original. The questionnaire investigated these aspects by proposing a seven-point scale for each dimension of the perception.

Figure 3 presents the outcomes of the perception of craft beer with glass packaging vs. can packaging in a spider-web graph. Respondents declared cans to be perceived as more youthful, more sophisticated, more global, more original, and trendier than glass bottles. It is also considered to be cheaper. Conversely, beer bottles were perceived to be more traditional, conventional, more expensive, and suitable to emphasize aspects related to the production area. Lastly, regarding pleasure, naturalness, and social and emotional aspects, no relevant differences were identified for different packaging materials.

Aluminum packaging in the Italian craft beer sector represents a novelty. It is most appreciated in historical brewing countries because can packaging is lighter, more resistant, more fitting for logistic operations, and for its sustainable characteristics. In Italy, aluminum seems to be attractive to engage younger consumers through narrative packaging communication strategies.

### 4.4. Benchmarking with Actual Breweries' Communication Strategies

To compare consumer preference analyses with marketing and communication strategies proposed by producers and to verify whether the current marketing and communication strategies adopted by Italian breweries are coherent with new consumers' needs and wants, in this study, an explorative investigation on the communication strategies of Italian craft beer enterprises on social networks was carried out.

In particular, as described in Table 7, starting from data obtained by the most recent annual report, AssoBirra [4], which reports all active breweries in Italy, a sample of 139 enterprises stratified by geographical area was randomly extracted. In this sample, an analysis concerning key web-based communication strategies was conducted. In particular, four main aspects were examined: the number of monthly activities on Facebook and Instagram social networks, the presence of a company website, and active e-commerce endeavors.

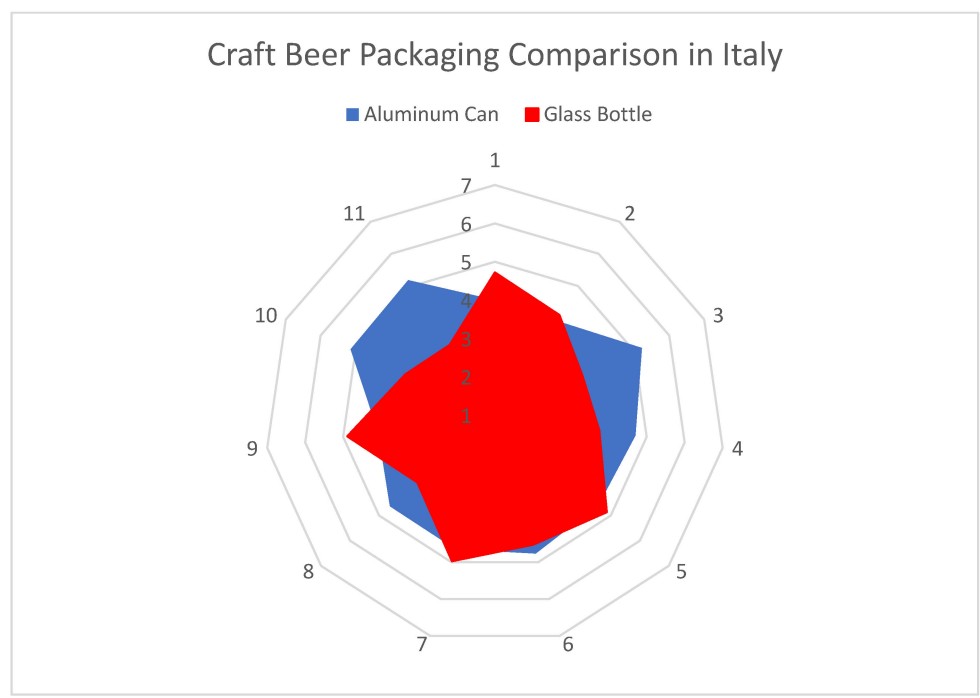

**Figure 3.** Comparison between Italian consumer perceptions of glass and can packaging for craft beer in (level of preference from 1 to 7).

**Table 7.** Sample of Italian craft breweries.

|  | Region | Number of breweries | Sample |
|---|---|---|---|
| Area 1 | Liguria | 34 | 3 |
|  | Lombardia | 245 | 26 |
|  | Piemonte | 120 | 12 |
|  | Valle d'aosta | 10 | 1 |
|  | Tot. | 409 | 42 |
| Area 2 | Emilia-romagna | 118 | 12 |
|  | Friuli venezia giulia | 51 | 5 |
|  | Trentino alto-adige | 55 | 6 |
|  | Veneto | 159 | 16 |
|  | Tot. | 383 | 39 |
| Area 3 | Lazio | 113 | 11 |
|  | Marche | 75 | 8 |
|  | Sardegna | 53 | 5 |
|  | Toscana | 118 | 12 |
|  | Umbria | 37 | 4 |
|  | Tot. | 396 | 29 |
| Area 4 | Abruzzo | 56 | 6 |
|  | Basilicata | 26 | 3 |
|  | Calabria | 37 | 4 |
|  | Campania | 99 | 10 |
|  | Molise | 15 | 2 |
|  | Puglia | 92 | 9 |
|  | Sicilia | 80 | 8 |
|  | Tot. | 405 | 29 |
| Italy | Tot. | 1593 | 139 |

Source: Own elaboration from Microbirrifici.org data [5].

As shown in Figure 3, the investigation identified that Italian craft beer producers mainly use the own website and Facebook to communicate with consumers; e-commerce was adopted by only 62% of the sample.

These outcomes can be better understood if they are linked to key concepts of marketing communication. In fact, based on the information sharing model called "One-To-Many", websites are deemed a traditional marketing communication tool. In fact, similarly to print, radio, and television, in this type of communication, a single source provides information, which is the same for everybody, to multiple receivers. In contrast, social media is considered as a marketing communication tool, and "Many-To-Many" and "Many-to-One" information sharing models. In these models, information is generated from multiple sources and is received by multiple sources or generated by multiple sources towards a single receiver, respectively. In particular, the Many-To-Many model is useful for active group discussions and to share opinions and ideas; communication based on the Many-to-One model is highly useful for enterprises to obtain feedback from their consumers. E-commerce is considered the most valuable tool based on the Many-to-One model because feedback from consumers is not only obtained through discussions or other enterprises' activities, but is highest because consumers genuinely buy company products [48–51].

Based on these concepts, the results of this investigation highlight that Italian craft beer producers principally use traditional communication tools and channel their focus on social media, generally preferring Facebook to Instagram.

Comparing current communication strategies adopted by producers with the results of the analysis of consumer preferences and perceptions, notably, these strategies are not fully consistent with the contemporary Italian consumer trend. In particular, an emerging consumer target was identified. This is characterized by young people regularly using social networks, with high knowledge and awareness of craft beer. For them, communication strategies based on more consumer engagement and participative marketing actions, as well as the increased use of Instagram, could be considered some simple suggestions to improve marketing communication [52–56].

In fact, as also shown in a study conducted by Statista in 2020, the key Instagram users are young people aged between 18 and 24 years old, whereas Facebook is preferred by older users. Consequently, for the emerging target consumers of craft beer in Italy, Instagram represents a more appropriate social media platform than Facebook, which, conversely, is mainly used by traditional and older craft beer consumers. For this consumer base, communications in ho.re.ca. are still considered a suitable and effective marketing strategy.

## 5. Conclusions

This paper can be considered as an attempt to analyze actual consumer preferences for craft beer through artificial intelligence and machine learning methodologies. The results were compared with actual communication strategies adopted by Italian breweries. The goal was to verify whether producers correctly choose their marketing and communication strategies, based on realistic consumers needs and wants.

In general, the results of this study highlight a dynamic market characterized by its two most relevant consumer clusters. One of these includes conservative and older consumers, who choose craft beer as an imperfect substitute for wine; the other one is an emerging segment composed of young people who, in craft beer consumption, reflect their modern and dynamic lifestyles.

Referring to products' characteristics, the analysis identified that taste and can packaging are the most appreciate attributes, whereas web-based and participatory communication, attractive packaging, and sales over the ho.re.ca channel can be considered as innovative marketing strategies for producers.

At present, only a small number of breweries use these innovative strategies, probably because they consider this new approach to be inappropriate for an "artisanal" product. In the Italian market, the preference for personal communication in the ho.re.ca channel by producers could be a limiting factor for further development of the sector. In fact, although this communication strategy may still be suitable to attract more traditional and older consumers, it appears to no longer be profitable considering characteristics of new consumer segments. Additionally, mixed strategies, aimed to attract mass markets,

do not seem suitable either. In fact, in Italy, the recent diffusion of brewing culture, the increase in producers, the high product differentiation, and the lack of in-depth knowledge of their specific attributes confuses many interested consumers who frequently make mistakes in their purchases. In these cases, effective communication strategies are desirable. Visits to breweries, informative campaigns using social media, and other participative marketing activities might be useful to increase knowledge and, consequently, improve consumers' choices.

In summary, in the Italian market, characterized by a high number of local and small-scale breweries offering different products, the results of this study suggest producers who intend to implement effective marketing strategies, to identify their consumer targets and clear strategic positioning, and in agreement, to choose the most appropriate activities.

Compared with other countries, where craft beer is considered a traditional beverage, the results also highlighted that the Italian craft beer industry is evolving as a stand-alone sector, separate from wine and industrial beer. For these specific market characteristics, competitive strategies, as adopted by international beer companies, are not always suitable for the Italian market.

In this study, communication on social media was considered an important aspect to understand Italian craft beer producers' propensity to employ innovative marketing and communication strategies. Future studies aiming to expand the knowledge of other marketing actions and tools could be useful for identifying additional strategies suitable to attract new consumers.

**Author Contributions:** Conceptualization, V.P., A.D.P. and A.S.; Literature review and background analysis, V.P. and A.D.P.; Methodology and statistical analysis, A.S.; Empirical analysis, V.P.; Data collection and curation, A.D.P.; Supervision, V.P.; Writing—original draft preparation, V.P., A.D.P. and A.S.; Writing—review and editing, V.P. and A.S. All authors have read and agreed to the published version of the manuscript.

**Funding:** This research received no external funding.

**Conflicts of Interest:** The authors declare no conflict of interest.

## Notes

1.  Nielsen divides every country into different areas. Italy is divided into four areas: area 1 (northwest area), area 2 (northeast area), area 3 (central area), and area 4 (south area).

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
