# Peer review of "Consumer Preferences for Craft Beer by Means of Artificial Intelligence: Are Italian Producers Doing Well?"

_beverages, doi:10.3390/beverages9010026_

Round 1

Reviewer 1 Report

Tittle:  CONSUMER PREFERENCES FOR CRAFT BEER BY MEANS 2 OF ARTIFICAL INTELLIGENCE: ARE ITALIAN 3 PRODUCERS DOING WELL?

The research topic is interesting. The introduction is relatively well written.

Some comments / recommendations from the reviewer.

2.2. The craft beer sector in Italy

no reference in the text to Figure 1

3.2. Data collection

Line 200-213, page 6: please describe in more detail the sections related to purchase behavior and consumption habits; the perception about two different packaging; motivations and attitudes of consumers (e.g. how many questions were in the sections, what questions, what scale was used, what attributes were used?)

4.1. Sample and data

no reference in the text to Table 3;  

page 308: As shown in Table 3 (should be Table 4)

4.2. Consumers’ profile and Willingness to Pay

Line 321-336, page 9,10: methodological issues. They should be moved

Line 340-341 What about the Figure? no reference in the text. Maybe delete?

Table 5 e.g. w_friends; without abbreviations

4.3. Benchmarking with actual breweries’ communication strategies

no reference in the text to Table 7

no title for Fig.4

Incorrect citations in the manuscript according to the requirements of the Journal

Author Response

Tittle:  CONSUMER PREFERENCES FOR CRAFT BEER BY MEANS 2 OF ARTIFICAL INTELLIGENCE: ARE ITALIAN 3 PRODUCERS DOING WELL?

Reviewer 2

The article CONSUMER PREFERENCES FOR CRAFT BEER BY MEANS 2 OF ARTIFICAL INTELLIGENCE: ARE ITALIAN 3 PRODUCERS DOING WELL? aims to analyze consumer preferences and estimated willingness to pay for craft beer compared to industrial products in Italy. The article is well written and the topic addressed is very interesting and relevant. However, it needs a little revision before being published. For example, on line 66 where de reads "been", it should read "beer". Authors also need to improve the resolution of figures. Therefore, I propose that, after reviewing the text and graphics, the article may be published.

Reviewer 1

The research topic is interesting. The introduction is relatively well written.

Some comments / recommendations from the reviewer.

2.2. The craft beer sector in Italy

no reference in the text to Figure 1 Reference in the text to Figure 1 has been added

3.2. Data collection

Line 200-213, page 6: please describe in more detail the sections related to purchase behavior and consumption habits; the perception about two different packaging; motivations and attitudes of consumers (e.g. how many questions were in the sections, what questions, what scale was used, what attributes were used?)

4.1. Sample and data 

no reference in the text to Table 3; 

page 308: As shown in Table 3 (should be Table 4)

In the text, there was confusion in the tables numbering. It has been corrected.

4.2. Consumers’ profile and Willingness to Pay

Line 321-336, page 9,10: methodological issues. They should be moved

Line 340-341 What about the Figure? no reference in the text. Maybe delete?

Table 5 e.g. w_friends; without abbreviations

Everything has been modified as suggested

4.3. Benchmarking with actual breweries’ communication strategies 

no reference in the text to Table 7 Reference in the text to Table 7 was added

no title for Fig.4 the title was added

Incorrect citations in the manuscript according to the requirements of the Journal

citations in the manuscript are now in according to the requirements of the Journal

  • References: References must be numbered in order of appearance in the text (including table captions and figure legends) and listed individually at the end of the manuscript. We recommend preparing the references with a bibliography software package, such as EndNoteReferenceManager or Zotero to avoid typing mistakes and duplicated references. We encourage citations to data, computer code and other citable research material. If available online, you may use reference style 9. below. 
  • Citations and References in Supplementary files are permitted provided that they also appear in the main text and in the reference list.

In the text, reference numbers should be placed in square brackets [ ], and placed before the punctuation; for example [1], [1–3] or [1,3]. For embedded citations in the text with pagination, use both parentheses and brackets to indicate the reference number and page numbers; for example [5] (p. 10). or [6] (pp. 101–105).

The reference list should include the full title, as recommended by the ACS style guide. Style files for Endnote and Zotero are available.

References should be described as follows, depending on the type of work:

  • Journal Articles:
    1. Author 1, A.B.; Author 2, C.D. Title of the article. Abbreviated Journal NameYearVolume, page range.
  • Books and Book Chapters:
    2. Author 1, A.; Author 2, B. Book Title, 3rd ed.; Publisher: Publisher Location, Country, Year; pp. 154–196.
    3. Author 1, A.; Author 2, B. Title of the chapter. In Book Title, 2nd ed.; Editor 1, A., Editor 2, B., Eds.; Publisher: Publisher Location, Country, Year; Volume 3, pp. 154–196.
  • Unpublished materials intended for publication:
    4. Author 1, A.B.; Author 2, C. Title of Unpublished Work (optional). Correspondence Affiliation, City, State, Country. year, status(manuscript in preparationto be submitted).
    5. Author 1, A.B.; Author 2, C. Title of Unpublished Work. Abbreviated Journal Name year, phrase indicating stage of publication (submittedacceptedin press).
  • Unpublished materials not intended for publication:
    6. Author 1, A.B. (Affiliation, City, State, Country); Author 2, C. (Affiliation, City, State, Country). Phase describing the material, year. (phase: Personal communication; Private communication; Unpublished work; etc.)
  • Conference Proceedings:
    7. Author 1, A.B.; Author 2, C.D.; Author 3, E.F. Title of Presentation. In Title of the Collected Work(if available), Proceedings of the Name of the Conference, Location of Conference, Country, Date of Conference; Editor 1, Editor 2, Eds. (if available); Publisher: City, Country, Year (if available); Abstract Number (optional), Pagination (optional).
  • Thesis:
    8. Author 1, A.B. Title of Thesis. Level of Thesis, Degree-Granting University, Location of University, Date of Completion.
  • Websites:
    9. Title of Site. Available online: URL (accessed on Day Month Year).
    Unlike published works, websites may change over time or disappear, so we encourage you create an archive of the cited website using a service such as WebCite. Archived websites should be cited using the link provided as follows:
    10. Title of Site. URL (archived on Day Month Year).

  1. Bowman, C.M.; Landee, F.A.; Reslock, M.A. Chemically Oriented Storage and Retrieval System. 1. Storage and Verification of Structural Information. J. Chem. Doc. 19677, 43-47; DOI:10.1021/c160024a013.
  • References to books should cite the author(s), title, publisher, publisher location (city and country), publication year, and page:
  1. Smith, A.B. Textbook of Organic Chemistry; D. C. Jones: New York, NY, USA, 1961; pp 123-126.
  • In referring to a book written by various contributors, cite author(s) first:
  1. Winstein, S.; Henderson, R.B. In Heterocyclic Compounds; Elderfield, R.C., Ed.; Wiley: New York, NY, USA, 1950; Vol. 1, Chapter 1, p 60.

  • Research manuscripts should comprise:
    • Front matter: Title, Author list, Affiliations, Abstract, Keywords. 
    • Research manuscript sections: Introduction, Materials and Methods, Results, Discussion, Conclusions (optional).
    • Back matter: Supplementary Materials, Acknowledgments, Author Contributions, Conflicts of Interest, References.

  • Keywords: Three to ten pertinent keywords need to be added after the abstract. We recommend that the keywords are specific to the article, yet reasonably common within the subject discipline.

For research articles with several authors, a short paragraph specifying their individual contributions must be provided. The following statements should be used "Conceptualization, X.X. and Y.Y.; Methodology, X.X.; Software, X.X.; Validation, X.X., Y.Y. and Z.Z.; Formal Analysis, X.X.; Investigation, X.X.; Resources, X.X.; Data Curation, X.X.; Writing – Original Draft Preparation, X.X.; Writing – Review & Editing, X.X.; Visualization, X.X.; Supervision, X.X.; Project Administration, X.X.; Funding Acquisition, Y.Y.”, 

Reviewer 2 Report

The article CONSUMER PREFERENCES FOR CRAFT BEER BY MEANS 2 OF ARTIFICAL INTELLIGENCE: ARE ITALIAN 3 PRODUCERS DOING WELL? aims to analyze consumer preferences and estimated willingness to pay for craft beer compared to industrial products in Italy. The article is well written and the topic addressed is very interesting and relevant. However, it needs a little revision before being published. For example, on line 66 where de reads "been", it should read "beer". Authors also need to improve the resolution of figures. Therefore, I propose that, after reviewing the text and graphics, the article may be published.

Author Response

(The authors gave the same response as above.)
